# Protective Effects of CISD2 and Influence of Curcumin on CISD2 Expression in Aged Animals and Inflammatory Cell Model

**DOI:** 10.3390/nu11030700

**Published:** 2019-03-25

**Authors:** Chai-Ching Lin, Tien-Huang Chiang, Yu-Yo Sun, Muh-Shi Lin

**Affiliations:** 1Department of Biotechnology and Animal Science, College of Bioresources, National Ilan University, Yilan 26047, Taiwan; lincc@niu.edu.tw (C.-C.L.); yellowstartree@gmail.com (T.-H.C.); 2Department of Pediatrics, Emory University School of Medicine, Atlanta, GA 30322, USA; ichpowersun@gmail.com; 3Division of Neurosurgery, Department of Surgery, Kuang Tien General Hospital, Taichung 43303, Taiwan; 4Department of Biotechnology, College of Medical and Health Care, Hung Kuang University, Taichung 43302, Taiwan; 5Department of Health Business Administration, College of Medical and Health Care, Hung Kuang University, Taichung 43302, Taiwan

**Keywords:** trauma, neurodegeneration, aging, curcumin, CISD2, CISD2-dependent manner

## Abstract

Background: Inflammation and mitochondrial dysfunction have been linked to trauma, neurodegeneration, and aging. Impairment of CISD2 expression may trigger the aforementioned pathological conditions in neural cells. We previously reported that curcumin attenuates the downregulation of CISD2 in animal models of spinal cord injury and lipopolysaccharide (LPS)-treated neuronal cells. In this study, we investigate (1) the role of *CISD2* and (2) how curcumin regulates CISD2 in the aging process. Materials and methods: The serial expression of CISD2 and the efficacy of curcumin treatment were evaluated in old (104 weeks) mice and long-term cultures of neural cells (35 days in vitro, *DIV*). LPS-challenged neural cells (with or without siCISD2 transfection) were used to verify the role of curcumin on CISD2 underlying mitochondrial dysfunction. Results: In the brain and spinal cord of mice aged P2, 8, 25, and 104 weeks, we observed a significant decrease in CISD2 expression with age. Curcumin treatment in vivo and in vitro was shown to upregulate CISD2 expression; attenuate inflammatory response in neural cells. Moreover, curcumin treatment elevated CISD2 expression levels and prevented mitochondrial dysfunction in LPS-challenged neural cells. The beneficial effects of curcumin in either non-stressed or LPS-challenged cells that underwent siCISD2 transfection were significantly lower than in respective groups of cells that underwent scrambled siRNA-transfection. Conclusions: We hypothesize that the protective effects of curcumin treatment in reducing cellular inflammation associated trauma, degenerative, and aging processes can be partially attributed to elevated CISD2 expression. We observed a reduction in the protective effects of curcumin against injury-induced inflammation and mitochondrial dysfunction in cells where CISD2 expression was reduced by siCISD2.

## 1. Introduction

Aging, neurodegenerative disease, and trauma are important health issues affecting the elderly. At present, there are more than 600 million people over the age of 65 [1], which represents roughly 10% of the global population. Inflammation [2,3,4] and mitochondrial dysfunction [5,6] are common to all of these pathological processes. Persistent low-grade inflammation has been associated with aging, aging-related diseases, and neurodegenerative diseases. Inflammation is often localized in the central nervous system (CNS) (i.e., neuroinflammation) or internal organs and has been linked to Alzheimer’s disease (AD), Parkinson’s disease (PD), multiple sclerosis (MS) [7,8,9,10,11,12], cardiovascular disease, type II diabetes, and osteoporosis [13]. Excessive inflammatory stimulation over extended periods can also lead to mitochondrial dysfunction. High-grade impact to the CNS, such as traumatic brain injuries (TBI), spinal cord injuries (SCI), and hemorrhagic/ischemic stroke [14], can lead to extensive alteration of mitochondrial dynamics, including an increase in membrane permeabilization, oxidative phosphorylation, and the production of mitochondrial reactive oxygen species (ROS) [15,16]. The same effects have been associated with prolonged inflammation driven by excessive glial activation [17]. Advanced mitochondrial dysfunction has also been shown to exacerbate inflammatory processes [18], resulting in neuronal damage and negative neurological outcomes. Thus, any treatment strategy that protects against inflammation and mitochondrial dysfunction could potentially have therapeutic benefits for any of these diseases.

Researchers have shown that the gene *CISD2* (CDGSH iron sulfur domain 2) has protective effects against inflammatory responses [19] and mitochondrial dysfunction-driven apoptosis [20]. *CISD2* detected in the outer membrane of mitochondria has also been shown to maintain mitochondrial integrity. Reports on *CISD2* knockout mice revealed that CISD2 deficiency induced mitochondrial dysfunction accompanied by cell death [20]. CISD2 has been shown to promote the binding of BCL2 to BECN1, which regulates cellular autophagy/apoptosis [21]. It is very likely that anti-inflammatory and/or anti-apoptotic therapies based on CISD2 could be used to reduce the effects of aging, neurodegenerative disease, and CNS trauma.

Curcumin is a herbal supplement derived from the root of turmeric (*Curcuma longa*), a member of the ginger family. From a structural perspective, curcumin can be categorized as a phenolic compound, which exerts antioxidant effects via the structure of hydroxyl groups and phenolic rings [22]. This natural compound has been popularized as a cooking spice; however, it has also been recognized for its protective effects against inflammation and oxidation. Curcumin has been shown to promote the decay of free radicals through electron donation from phenolic hydroxyl groups in ROS scavenging [23]. The effects of curcumin have been shown to protect against injury-triggered neuroinflammation. Curcumin attenuates a variety of injury-associated inflammatory mediators, such as pro-inflammatory transcription factor activator protein-1 (AP-1), nuclear factor κB (NFκB), and iNOS [24]. In a previous study, our research team discovered that curcumin attenuates injury-stimulated glial activation by decreasing the expression of glial fibrillary acid protein (GFAP) [25] and downregulating the secretion of injury-induced RANTES (Regulated on Activation, Normal T Cell Expressed and Secreted) in the astrocytes of rats following SCI [26]. The anti-inflammation and anti-oxidation effects of curcumin raises the possibility of its use in the treatment of aging [27], neurodegenerative diseases [28,29], and CNS injuries [25]. However, the molecular mechanisms underlying the protective effects of curcumin have yet to be fully elucidated.

Previous in vivo and in vitro studies revealed that the protective effects of curcumin after injuries involve the regulation of CISD2. Injuries led to the downregulation of CISD2 expression in lipopolysaccharide (LPS)-challenged astrocytes and rats following SCI. The administration of curcumin attenuated the reduction of CISD2 expression after injury in vivo and in vitro [19]. However, the role of curcumin on CISD2 in aging process remains unclear. In the current study, we sought to delineate the roles of CISD2 in non-stressed and advanced inflammation associated with aging, CNS injury, and neurodegenerative disease. We also examined the means by which curcumin regulates the expression of CISD2 in LPS-treated neural cells and the in vitro model of aging. Despite numerous reports on the benefits of curcumin, no previous study has reported on the CISD2-elevating effect or its association with the anti-inflammatory and anti-oxidative properties of this compound.

## 2. Materials and Methods

### 2.1. Chemicals and Reagents

Curcumin was dissolved in DMSO (both from Sigma-Aldrich, St. Louis, MO, USA). LPS was obtained from *Escherichia coli* serotype 055: B5, L-2880 (Sigma Chemical Co., St. Louis, MO, USA). U0126 was purchased from Sigma-Aldrich (St. Louis, MO, USA). AG490 and RO-318220 were obtained from Calbiochem (San Diego, CA, USA). LY294002 was purchased from Calbiochem (Cambridge, MA, USA). alamarBlue^®^ Cell Viability Assay (Catalog: 88951) was obtained from Life Technologies (Carlsbad, CA, USA). The apparatus and materials for flow cytometry were purchased from Life Technologies (Carlsbad, CA, USA). We also employed the MitoProbe™ JC-1 Assay Kit (Catalog: M34152), CellROX^®^ Deep Red Flow Cytometry Assay Kit (Catalog: C10491), Chromatin Condensation & Membrane Permeability Dead Cell Apoptosis Kit/YO-PRO™-1, and PI dyes (Catalog: V23201).

### 2.2. Animals

Wild-type C57BL/6JNarl mice weighing 22–28 g were purchased from the National Laboratory Animal Centre (NLAC, Taipei, Taiwan) and kept at five mice per cage for least five days following their arrival at our laboratory. The animals were provided unlimited access to food and water and were maintained under a 12:12 h dark–light cycle. This study was performed in accordance with the guidelines outlined by the Experimental Animal Laboratory and approved by the Animal Care and Use Committee at National Ilan University, Yilan, Taiwan.

### 2.3. Curcumin Treatment of Mice

Mice were randomly divided into two groups: controls and curcumin-treated (each group n = 4). The animals were anesthetized with isoflurane. The control group was intraperitoneally administered 10% DMSO, and the curcumin-treated group was administered a single injection of curcumin intraperitoneally at a concentration of 40 mg/kg. In previous studies, this dosage has been shown to bestow neuroprotective effects [25,26]. Animals were allowed to recover on a heating pad at 36.5 °C. Subcutaneously, 1.0 mL saline was administered for rehydration. The animals were eating and drinking within 3 h after procedure. For all cases (i.e., with or without curcumin treatment), spinal cord tissue was harvested. Whole cell lysate from tissue was used for western blot analysis.

### 2.4. Cells

Human undifferentiated neuroblastoma cell line SH-SY5Y was purchased from ATCC (Manassas, VA, USA). Cells were grown in 1:1 mixture of Ham’s F12 nutrient and Dulbecco’s modified Eagle’s medium (DMEM) supplemented with 10% foetal bovine serum (FBS), 1% non-essential amino acid solution, penicillin (100 U/mL) and streptomycin (100 mg/mL). The cells were maintained under 5% CO_2_ at 37 °C in a humidified atmosphere. Non-stressed SH-SY5Y (treated or untreated with curcumin) were conducted to perform mRNA analysis, and cell viability. LPS-challenged cells with or without curcumin were tested with siRNA knockdown, flow cytometry, and cell viability.

As described in a previous study [26], we used primary astrocyte cultures for the experiment on signaling pathways associated with curcumin-induced CISD2 mRNA expression. The cerebral cortex was harvested from 1- to 2-day-old Sprague–Dawley (SD) rats. At 6–8 days after cell seeding, the suspended cells were removed to obtain the layer of pure astrocytes that adhered to the bottom of the culture flasks. The morphology indicated that the astrocyte culture was of at least 85% purity.

Long-term primary culture of astrocyte, an “age in the dish” model, could serve as a model to mimic conditions of aging process [30,31]. Primary cultured astrocytes were grown for 35 *DIV* prior to experimentation. Cells were treated with 10 μM/L cytosine arabinoside to limit astrocyte division and to maintain the purity of the cultures. We used astrocytes generated from rat fetal neural stem cells (Invitrogen, Carlsbad, CA, USA) in accordance with the culture protocol described in the product guidelines and the literature [32,33]. Briefly, rat fetal neural stem cells were isolated from the brain cortex of fetal SD rats at day 14 of gestation. The cells were expanded in 90% N-2/DMEM/F-12 medium and 10% DMSO.N-2/DMEM/F-12 medium with 10 ng/mL basic fibroblast growth factor. Rat fetal neural stem cells spontaneously differentiated into astrocytes following the withdrawal of growth factors from the cell culture.

To induce astrocyte differentiation, StemPro^®^ NSC SFM (Invitrogen, Carlsbad, CA, USA) was used without epidermal growth factors or basic fibroblast growth factors in accordance with the culture protocol described in the product guidelines. GFAP staining was used to verify that the astrocytes were of at least 90% purity. 

### 2.5. Curcumin Treatment of Lipopolysaccharide (LPS)-Challenged Neural Cells

SH-SY5Y cultured cells were challenged with LPS. To study the effects of curcumin on cells that sustained injuries, neural cell cultures (1 × 10^6^ in a 35-mm dish) were incubated without LPS or curcumin (control group), with 1 μg/mL LPS, or with 1 μg/mL LPS and 1 μM curcumin for 24 h. In previous studies, the dosage of LPS and curcumin has been shown to bestow inflammation-induced [19,26] and optimal neuroprotective effects [19,25,26,30], respectively. Each experiment was performed at least twice using at least three different astrocyte cultures.

### 2.6. Reverse-Transcription Polymerase Chain Reaction and Real-Time Quantitative Reverse-Transcription Polymerase Chain Reaction (qRT-PCR)

Total RNA was prepared by directly lysing the cultured cells in extraction buffer (Trizol/phenol/chloroform) and reverse transcribing the mRNA into cDNA using oligo-dT and SuperScript II reverse transcriptase (Invitrogen, Carlsbad, CA, USA). The cDNAs were subjected to polymerse chain reaction (PCR) to measure the expression of the genes of interest, and the housekeeping gene cyclophilin was used as an internal control. The PCR protocol included 25 cycles of denaturation for 1 min at 94 °C, annealing for 1 min at 55 to 60 °C, and extension for 1 min at 72 °C. The primers used for PCR are listed in Table 1. For real-time quantitative reverse-transcription polymerase chain reaction (qRT-PCR), the cDNA samples were analyzed using the SYBR Green gene expression system (ABI PRISM 7300 HT real-time PCR system; Applied Biosystems, Foster City, CA, USA). Minor groove binding dyes and primers for the detection of the genes of interest and cyclophilin were designed by ABI. The threshold cycle (Ct) (the fractional number of cycles at which the amount of amplified target reached a fixed threshold) was determined and the Ct value of targeted genes was normalized using cyclophilin as an internal control. Each measurement was performed at least three times.

### 2.7. RNA Interference and Transfection

We used small interfering RNA (siRNA) specific to CISD2 mRNA to knockdown the expression of CISD2 in cultured microglia. Cells were transfected using a set of siRNAs specific to CISD2 or scrambled RNA (Silencer^®^ Pre-designed siRNA, Ambion, Austin, TX, USA) using Lipofectamine™ 2000 reagent (Invitrogen, Carlsbad, CA, USA). Targeted SiRNA-CISD2 sequences were 5′-GUCCUCUCAUCCUGAAGAATT-3′ and 5′-UUCUUCAGGAUGAGAGGACTT-3′. Five hours after transfection, the Lipofectamine 2000-containing medium was replaced with microglial culture medium to allow the cells to recover for another 67 h. Real-time qRT-PCR for CISD2 mRNA was performed to verify knockdown efficiency.

### 2.8. Immunoblotting of CISD2

Total protein was extracted from cultured neuron and glial cells in a lysis buffer containing 20 mM Tris-HCl, 0.1% SDS, 0.8% NaCl, and 1% Triton-X 100. Electrophoresis with a 12% gradient was performed to separate the protein extracts. Proteins that underwent electrophoresis were electro-transferred to a nitrocellulose membrane, and then incubated with blocking reagent, primary antibodies (anti-CISD2 (1:500) (Thermo Scientific, Waltham, MA, USA, PA5-34545), and anti-GAPDH (1:500) (Millipore, Billerica, MA, USA)) for 12 h at 4 °C, before being washed and incubated with anti-rabbit IgG (HRP-conjugated secondary antibody) (Merck Millipore, Cat.#12-348) for 1 h. Results were detected using chemiluminescence (Merck Millipore, WBKLS0500). Bands of interest were visualized and quantified using the ImageQuant^TM^ LAS 4000 (GE Healthcare Life Sciences, Marlborough, MA, USA).

### 2.9. Chemicals and Fluorescent Dyes for the Study of Mitochondrial Function

Mitochondrial function was studied using flow cytometry analysis using specific fluorescent probes to detect changes in mitochondrial membrane potential [DeltaPsi(m)], ROS formation, and cellular apoptosis. Changes in DeltaPsi(m) were detected using the ratiometric indicator 5,5′,6,6′-tetra-chloro-1,1′,3,3′-tetraethylbenzimidazolylcarbocyanine iodide (JC-1) (MitoProbe™ JC-1 Assay Kit). Low DeltaPsi(m) levels (appearing as monomer green JC-1 fluorescence) indicated impending cellular apoptosis, unlike the control cells with high DeltaPsi(m) (appearing as j-aggregated red fluorescence). The ratio of PE+ (j-aggregated red fluorescence) AF488+ (monomer green fluorescence) to PE-AF488+ (simplified as PE+ to PE−) was used to represent the extent of cellular apoptosis. ROS formation was investigated using CellROX^®^ Deep Red reagent. Cellular apoptosis was analyzed by loading the fluorescent dyes, YO-PRO-1 and PI. All of the aforementioned fluorescent probes were stained at room temperature for 20–30 min. After dye loading, the cells were rinsed three times using phosphate-buffered saline (PBS). Flow cytometry was performed as described in [34]. Data acquisition and analysis were performed using a flow cytometer (LSRFortessa Cell Analyzer, Becton-Dickinson, Franklin Lakes, NJ, USA). Flow cytometry data were analyzed using FlowJo software Version 10 (TreeStar Inc., Ashland, OR, USA).

### 2.10. Cell Viability

The survival rate of cultured neural cells was measured using the alamarBlue ^TM^ assay (Life Technologies, Carlsbad, CA, USA) according to the manufacturer’s instructions. Briefly, cells (10^4^ cells/100 μL medium) were plated in a 96-well microplate 24 h before use. alamarBlue ^TM^ solution was added at an amount equal to 10% of the culture volume. Cells were kept at 37 °C in an incubator environment of 5% carbon dioxide for 24 h. The fluorescence of the assay solution was then measured (excitation at 570 nm, emission at 600 nm) with a Thermo MultiSkan GO microplate reader (Thermo Scientific, Waltham, MA, USA).

### 2.11. Statistical Analysis

Independent two-sample *t* tests were used to compare data between the experimental groups. One-way analysis of variance (ANOVA) was used in cases where there were more than two groups. Statistical analysis was performed using GraphPad Prism software 5.0 (GraphPad Software Inc., La Jolla, CA, USA).

## 3. Results

### 3.1. Aging-Driven Decline in *CISD2* Expression in Naturally Aged Mice and Curcumin Protected against Aging-Driven Decline in *CISD2* Expression as Well as Aging-Augmented Inflammatory Responses in Long-Term Cultures of Neural Cells

A variety of molecular pathways underlie the pathologies of aging in the CNS and aging-related diseases such as chronic inflammation [14,35], mitochondrial dysfunction [5,6], and dysregulation of apoptosis [36]. The longevity gene, *CISD2*, has been shown to bestow protective effects, including anti-inflammatory effects [19] and the mitochondrial regulation of apoptosis [20]. This could be interpreted as CISD2 deficiency triggering an inflammatory cascade leading to mitochondrial dysfunction. Real-time qRT-PCR was used to examine CISD2 mRNA expression in the brain and spinal cord of mice at postnatal day 2, week 8, 25, and 104. Each experiment was performed at least three times. Results revealed a statistically significant decrease in the CISD2 mRNA levels with an increase in the age of the mouse brain (Figure 1A) and spinal cord (Figure 1B). These results suggest that *CISD2* inactivity and enhanced inflammatory responses may be the primary mechanisms underlying aging in the brain and spinal cord.

We also sought to characterize the efficacy of curcumin in providing neural protection against the effects of aging. This was achieved by examining the pharmacological effects of curcumin in astrocytes that underwent culturing over extended periods mimicking the conditions of aging. Astrocytes at 7 and 35 *DIV* were treated with 1 μM curcumin and then collected after a period of 24 h. Real-time qRT-PCR revealed that cells that underwent extreme aging presented lower CISD2 (*p* < 0.05, Figure 1C) as well as higher mRNA expression levels of iNOS (*p* < 0.05, Figure 1D) and RANTES (*p* < 0.001, Figure 1E), compared to the 7 *DIV* cells. Moreover, curcumin-treated cells presented a marked increase in CISD2 mRNA expression (*p* < 0.05, *p* < 0.01, respectively, Figure 1C) and attenuated aging-driven inflammation [iNOS (*p* < 0.01, *p* < 0.01, respectively, Figure 1D); RANTES (*p* < 0.001, *p* < 0.001, respectively, Figure 1E)], compared to cells that did not undergo curcumin treatment. Thus, in vivo and in vitro analyses indicate that a reduction in CISD2 promotes aging-related inflammation; curcumin rescued aging-associated signaling, perhaps through the upregulation of CISD2 expression and the attenuation of aging-related inflammation.

### 3.2. Knockdown of *CISD2* Expression in Neural Cells Led to High No Production with a Strong Inflammatory Reaction, Enhanced Apoptosis, and Attenuated Cell Survival

Based on our observations that the aging process is associated with a decrease in CISD2 expression and an increase in the production of inflammatory mediators, we sought to determine the physiological role of *CISD2* and its association with neural survival. We performed a loss-of-function study to attenuate CISD2 expression levels in order to verify a causal relationship between CISD2 expression and neural protection. We used SH-SY5Y neural cells, which are commonly used for the in vitro modeling of neuronal function associated with aging and neurodegenerative disease [37]. siRNA was used to knockdown CISD2 expression (siCISD2) in neural cells. Real-time qRT-PCR analysis revealed that the knockdown efficiency of siCISD2 in cultured SH-SY5Y cells was approximately 50%, compared to the scrambled RNA-transfected control group (Figure 2A). In real-time qRT-PCR analysis, siCISD2-transfected cells presented a marked increase in the mRNA expression of iNOS (*p* < 0.001, Figure 2B) and RANTES (*p* < 0.001, Figure 2C), compared to scrambled RNA-transfected cells. Conversely, real-time qRT-PCR analysis revealed a significant reduction in BCL2 mRNA expression in the siCISD2-transfected group, compared to the control group that underwent scrambled RNA-transfection (*p* < 0.05, Figure 2D). Furthermore, we observed a significant reduction in cell viability in siCISD2-transfected cells, compared to the group that underwent scrambled RNA-transfection (91.6 ± 2.1% vs. 100 ± 5.3%, *p* < 0.05) (Figure 2E). These in vitro findings indicate that CISD2 deficiency in neural cells that underwent siCISD2 transfection resulted in inflammatory responses, apoptosis, and impaired cell survival.

### 3.3. Curcumin Enhanced CISD2 mRNA and Protein Expression In Vivo and In Vitro via JAK/STAT Signaling Pathways

We first demonstrated in vivo that CISD2 is upregulated in mice treated with curcumin. Mice underwent intraperitoneal administration of normal saline or curcumin at a concentration of 40 mg/kg once per day for a period of two days (i.e., a total of two injections).

Western blot analysis was used to examine CISD2 protein expression in mouse spinal cords. Each experiment was performed three times. Mice treated with curcumin presented a significant increase in CISD2 protein expression in the spinal cord (*p* < 0.01), compared to untreated mice (Figure 3A).

The in vitro study used cultured astrocytes and neuron-like SH-SY5Y cell culture. This cell culture model was designed to mimic astroglia and neurons in the CNS, under the effects of aging and neurodegenerative disease. To evaluate the effects of curcumin on CISD2 expression, we treated two groups of cultured cells with 1 μΜ curcumin. Real-time qRT-PCR analysis revealed that treatment with curcumin led to an increase in CISD2 mRNA expression in cultured SH-SY5Y cells, compared to controls (*p* < 0.01, Figure 3B) and astrocytes (*p* < 0.05, Figure 3D). Conversely, cells treated with 1 μM curcumin presented a marked decrease in iNOS mRNA expression in neuron-like cells (*p* < 0.05, Figure 3C) and astrocytes (*p* < 0.01, Figure 3E), compared to cells that did not undergo curcumin treatment. We then sought to determine whether the increase in CISD2 mRNA expression in astrocytes induced by 1 μΜ curcumin involved the JAK/STAT, PI-3K, PKC, or MAPK pathways. We discovered that CISD2 mRNA expression was significantly downregulated following the inhibition of the JAK/STAT signaling pathways (*p* < 0.01, Figure 3F).

### 3.4. Curcumin Enhanced an LPS-Triggered Reduction in Mitochondrial Membrane Potential In Vitro

As mentioned in the literature, *CISD2* expression levels are significantly reduced by the aging process [20] or CNS injury [19]. A reduction in the expression of CISD2 (coding for a protein in the outer membrane of mitochondria) can undermine mitochondrial integrity and thereby impair mitochondrial membrane potential DeltaPsi(m). Having determined that curcumin attenuates pathology-driven CISD2 downregulation, we sought to determine whether curcumin affects changes in DeltaPsi(m) in neural cells that have sustained high-grade insults. Thus, we challenged SH-SY5Y neural cells with LPS to induce advanced inflammation and mitochondrial dysfunction in a manner that mimics CNS injuries and neurodegenerative diseases. We used siCISD2 to knockdown *CISD2* expression in LPS-challenged neural cells (treated with or without curcumin) to characterize the relationship between CISD2 and curcumin treatment. JC-1 dye was used to stain siCISD2-transfected and scrambled RNA-transfected LPS-challenged cells (treated with or without curcumin) followed by flow cytometry. The extent of cellular apoptosis in all groups was evaluated, as shown in Figure 4A.

DeltaPsi(m) levels in LPS-challenged groups were lower than those in the control groups (i.e., untreated with LPS), as follows: scrambled RNA-transfected (*p* < 0.01, labeled as **, Figure 4B) and siCISD2-transfected cells (*p* < 0.05, labeled as #, Figure 4B). LPS-challenged cells that were treated with 1 μM curcumin exhibited DeltaPsi(m) values exceeding those of LPS-challenged cells that were not treated with curcumin for scrambled RNA-transfected cells (*p* < 0.001, labeled as ***) and siCISD2-transfected cells (*p* < 0.001, labeled as ###) (Figure 4B). These data illustrate the efficacy of curcumin in enhancing injury-attenuated DeltaPsi(m)s.

### 3.5. Curcumin Protected against LPS-Augmented Reactive Oxygen Species (ROS) Formation In Vitro

CNS insults and neurodegeneration can lead to oxidative stress, resulting in the accumulation of ROS and mitochondrial degradation. ROS-driven mitochondrial dysfunction can further induce subsequent inflammatory reactions, resulting in a vicious pathological cycle of inflammation and mitochondrial dysfunction [38]. Thus, we sought to determine whether curcumin could protect against LPS-induced increases in ROS. SH-SY5Y neural cells stained with CellROX^®^ Deep Red reagent were analyzed using flow cytometry to indicate the extent of ROS formation (Figure 5A). We determined that ROS formation was higher in LPS-challenged neural cells than in control cells (i.e., without LPS treatment), as follows: scrambled RNA-transfected (*p* < 0.01, labeled as **) and siCISD2-transfected groups (*p* < 0.01, labeled as ###) (Figure 5B). Among LPS-challenged cells, those that underwent curcumin treatment exhibited ROS levels lower than those without curcumin treatment, as follows: scrambled RNA-transfected (*p* < 0.01, labeled as **) and siCISD2-transfected groups (*p* < 0.001, labeled as ###) (Figure 5B). These findings demonstrate the efficacy of curcumin in scavenging ROS due to injury. Moreover, a significantly higher percentage of dead cells were found in siCISD2-transfected groups, compared to scrambled RNA-transfected ones, as follows: control (27.5% vs. 5.92%) and LPS-challenged groups (23.6% vs. 11.2%) (Figure 4A). We confirm that CISD2 deficiency in neural cells is associated with high rates of cell death under non-stressed and injury-challenged status.

### 3.6. Curcumin Inhibited LPS-Induced Apoptosis In Vitro

Inflammation and/or mitochondrial dysfunction can lead to neural apoptosis and subsequent neurological deficit. Thus, we examined the anti-apoptotic effects of curcumin in injury-challenged neural cells. SH-SY5Y neural cells were labeled using YO-PRO-1 (as a marker for apoptosis) and PI (as a marker for necrotic cells). Flow cytometry was then used to indicate the extent of early cellular apoptosis (labeled as A, Figure 6A).

We observed a significant increase in apoptosis in the LPS-challenged group, compared to the control group (untreated with LPS), as follows: scrambled RNA-transfected (*p* < 0.05, labeled as *) and siCISD2-transfected groups of cells (*p* < 0.001, labeled as ###) (Figure 6B). Apoptosis was far less pronounced in LPS-challenged cells that underwent curcumin treatment, compared to LPS-challenged cells that did not undergo curcumin treatment, as follows: scrambled RNA-transfected cells (*p* < 0.05, labeled as *) and siCISD2-transfected cells (*p* < 0.05, labelled as #) (Figure 6B). Taken together, this provides evidence to support the assertion that curcumin protects against mitochondrial dysfunction and the corresponding inflammation by increasing injury-attenuated DeltaPsi(m)s, attenuating injury-augmented ROS formation, and decreasing injury-triggered cellular apoptosis in cells that underwent insult, such as those associated with CNS injury and neurodegenerative disease.

### 3.7. Curcumin Enhanced Cell Survival in Non-Stressed and Injury-Challenged Neural Cells

Non-stressed neural cells incubated with curcumin presented cell survival rates higher than those of cells that were not treated with curcumin, as follows: scrambled RNA-transfected group (*p* < 0.05, labeled as *) and siCISD2-transfected group (*p* < 0.05, labeled as #) (Figure 7). Under an injured microenvironment, the viability of LPS-challenged cells was significantly lower than that of the control group (untreated with LPS), as follows: scrambled RNA-transfected group (*p* < 0.001, labeled as ***) and siCISD2-transfected group (*p* < 0.001, labeled as ###) (Figure 7). In a sense, the administration of curcumin rescued many cells from death. The viability of LPS-challenged cells treated with 1 μM curcumin was markedly higher than that of LPS-challenged cells that were not treated with curcumin, as follows: scrambled RNA-transfected group (*p* < 0.01, labeled as **) and siCISD2-transfected group (*p* < 0.05, labeled as #) (Figure 7). These findings clearly demonstrate the beneficial effects of curcumin in promoting cell survival in non-stressed as well as injury-challenged neural cells. This suggests that perhaps curcumin could be applied to the treatment of aging, TBI, SCI, and neurodegenerative diseases.

### 3.8. Attenuated Survival Enhancing Effect of Curcumin in Non-Stressed as Well as LPS-Challenged Neural Cells

In evaluating the protective effects of curcumin on non-stressed and LPS-challenged neural cells (with or without siCISD2 transfection), we found that the cell viability of siCISD2 transfected cells (non-stressed cells, and injury-induced cells) was significantly lower after curcumin treatment, compared to non-stressed cells [*p* < 0.05 (labeled as &), Figure 7] and LPS-challenged cells [*p* < 0.01 (labeled as &&), Figure 7] in respective scrambled RNA-transfected control groups.

The protection afforded by curcumin (by promoting survival or combating inflammation and mitochondrial dysfunction) is less pronounced in cases of CISD2 deficiency. CISD2 clearly plays a critical role in the therapeutic effects of curcumin.

## 4. Discussion

The precise physiological function of *CISD2* has not been fully elucidated. In the current research, we used a siRNA gene attenuation model with *CISD2* knockdown to infer the physiological status of *CISD2*. Under non-stress conditions, CISD2 plays a protective role against inflammation and mitochondrial dysfunction. Thus, CISD2 deficiency can have broad detrimental effects. In this study, we found that the control neural cells (cells that underwent siCISD2-transfection but were not treated with LPS or curcumin) presented more extensive mitochondrial dysfunction, including decreased DeltaPsi(m)s (*p* < 0.05, Figure 4B, labeled as &), higher ROS levels (*p* < 0.05, Figure 5B, labeled as &), and more extensive cellular apoptosis (*p* < 0.05, Figure 6B, labeled as &), compared to control cells (cells that underwent scrambled RNA transfection but were not treated with LPS or curcumin). Therefore, we posit that CISD2 deficiency (as a proxy for aging) significantly increased iNOS expression, enhanced inflammatory responses, promoted mitochondrial dysfunction, attenuated anti-apoptotic *BCL2* gene expression, and eventually reduced neural cell viability.

In cases of advanced cellular insult (e.g., acute CNS trauma or neurodegenerative disease), CISD2 has been shown to influence the mechanisms underlying inflammation and mitochondrial dysfunction. CISD2 expression tends to be attenuated in cells that have sustained injury. The downregulation of CISD2 levels was demonstrated in rats with SCI and LPS-treated astrocytes [19]. LPS-induced inflammatory responses [25,26] and mitochondrial dysfunction (including impaired DeltaPsi(m) and elevated ROS levels) were observed in a previous study [26] as well as this current research. Specifically, in LPS-challenged *CISD2* knockdown cells, we reported that a CISD2 deficiency can result in elevated iNOS levels (in SH-SY5Y) [19], and enhanced inflammatory responses (in microglia, unpublished data). In the current research, neural cells were stimulated using LPS to trigger cellular injury, and then some of the cells were transfected with siCISD2. Compared to scrambled RNA-transfected cells, cells that underwent siCISD2-transfection exhibited more pronounced mitochondrial dysfunction, including lower DeltaPsi(m) values (*p* < 0.05, labeled as &, Figure 4), higher ROS levels (*p* < 0.01, labeled as &&, Figure 5), more extensive cellular apoptosis (*p* < 0.001, labeled as &&, Figure 6), and a reduction in cell survival (*p* < 0.05, labeled as &, Figure 7). Thus, we postulate that injury-attenuated *CISD2* expression resulted in inflammatory responses, mitochondrial dysfunction, including lower DeltaPsi(m) values, higher ROS levels, more extensive cellular apoptosis, and a reduction in cell survival, which could be expected to result in neurological deficit.

CISD2 has been linked to the upstream control of BCL2 and NFκB [39]. In response to stress, CISD2 can bind to BCL2 to enhance BCL2-BECN1 interactions, thereby antagonizing the apoptotic effects of Beclin 1 [40]. Moreover, NFκB exerts biphasic effects to promote or dissociate the Beclin 1/Bcl2 complex in response to microenvironmental changes [41]. In the current research, we demonstrated that CISD2 could significantly influence non-stressed and injury-challenged microenvironments. We postulate that the protective effects of CISD2 operate via upstream gene regulation, which would explain why CISD2 deficiency can have such a broad range of detrimental effects.

The curry spice, curcumin, is a natural component of turmeric, which is widely used in Indian dishes. The effects of curcumin have been demonstrated to result from its anti-inflammatory, anti-oxidizing, and immunomodulatory properties, which have proven beneficial in the treatment of inflammation and neurodegenerative conditions, such as aging [42], AD, PD, MS [43], TBI [44], and SCI [25]. It has been recently demonstrated that curcumin antagonizes the activation of NFκB and inhibits the production of TNFα, IL1β, and iNOS in various inflammatory diseases [45]. Curcumin presents anti-oxidant effects by decreasing lipid peroxidation and scavenging NO, which attenuates mitochondrial dysfunction [46]. The benefits of curcumin can be attributed to the regulation of multiple important molecular targets, including transcription factors, enzymes, cell cycle proteins, receptors, and cell surface adhesion molecules [47]. We previously found that treatment with curcumin could attenuate the downregulation of CISD2 expression in mice with SCI and LPS-challenged astrocytes [19], while attenuating injury-induced inflammation [25].

In the current study, we discovered that the pharmacological effects of curcumin involve the upregulation of CISD2 in non-stressed cells as well as injured cells (i.e., cells in which CISD2 is pathologically attenuated). Data obtained in this study show that treatment with curcumin can attenuate injury-induced mitochondrial dysfunction, decrease cellular apoptosis, and enhance cell survival. The increase in CISD2 may explain the benefits of curcumin on pathological conditions involving CISD2 deficiency and accompanying inflammation and mitochondrial dysfunction. We therefore postulate that the unique function of curcumin is its ability to elevate injury-attenuated CISD2 expression, thereby protecting against inflammation, mitochondrial dysfunction, and apoptosis during the pathogenesis of TBI, SCI, and neurodegenerative diseases. The beneficial effect of aging-CISD2-curcumin axis provides strong rationale for further development of curcumin-based treatments for patients suffering the aforementioned disorders. Nonetheless, the mechanism underlying the effect of curcumin on neuroinflammation and mitochondrial dysfunction will have to be verified using in vivo animal studies, such as ultrastructural and morphological analysis.

We found compromised protective properties against injury-induced mitochondrial dysfunction of curcumin in LPS-challenged neural cells in cases of CISD2 deficiency. In a comparison of LPS-challenged, and curcumin-treated cells (with or without siCISD2-transfection), cells that underwent siCISD2-transfection exhibited lower DeltaPsi(m)s [*p* < 0.001 (labeled as &&&) Figure 4], higher ROS levels [*p* < 0.05, (labeled as &) Figure 5] and higher rates of apoptosis [*p* < 0.001, (labeled as &&&) Figure 6], compared to scrambled RNA-transfected cells. CISD2 deficiency clearly influences the therapeutic effects of curcumin. The mechanism involved in the regulation of curcumin influencing CISD2 remains unknown. The research results known so far are as follows. CISD2 has been shown to mediate upstream regulation of inflammation and mitochondrial dysfunction-driven apoptosis. Similarly, curcumin protects against the above-mentioned pathological conditions. Data obtained in this study show that curcumin’s protective effects against inflammation and mitochondrial dysfunction can be attenuated in cases of CISD2 deficiency. Thus, we postulate that curcumin exhibits anti-inflammatory and anti-apoptotic properties via the action of *CISD2*. Moreover, we found that CISD2 can be pathologically attenuated in non-stressed and injured status and curcumin prevents this downregulation. Curcumin has been shown to preserve mitochondrial integrity and functions [48]. Therefore, the CISD2-elevating effect of curcumin would be likely due to prevent the loss of CISD2 expressing cells or direct effects of curcumin targeting CISD2. The detailed mechanism needs the further study.

Some considerations have been raised about the current curcumin research. First, it has been demonstrated that curcumin can cross the blood–brain barrier [43], and percolate into the cerebrospinal fluid (CSF) [49]. The CSF can circulate in the subarachnoid space around the brain and spinal cord, which achieves the delivery of curcumin to the spinal cord. Through this route, curcumin can exert protective effects in the CNS. In comparison with biodistribution of curcumin administered orally and intraperitoneally, a mouse study has shown the curcumin concentration in plasma is much lower by oral administration than by intraperitoneal injection [50]. The bioavailability of oral curcumin is low and advanced methods have been developed to increase the bioavailability of curcumin (e.g., adjuvants, nanoparticles, or structural analogues) [50]. In the current study, intraperitoneal administration of curcumin was performed to determine the therapeutic window and to assess the therapeutic effect of curcumin on CISD2 expression in an animal model. We found that curcumin protected against aging-driven CISD2 decline and aging-augmented inflammatory responses in vivo and in vitro. Oral administration or advanced methods will be considered in a further preclinical study. Second, as shown in Figure 7, SH-SY5Y neural cells incubated with curcumin alone presented cell survival rates higher than those of control cells in scrambled RNA-transfected (*p* < 0.05) and siCISD2-transfected groups (*p* < 0.05). Similarly, we found that PE+/PE− ratios of scrambled and siCISD2 (LPS+Curcumin+) cells were higher than respective (LPS−Curcumin−) control ones (Figure 4B). Therefore, the protective effect of curcumin on anti-apoptosis or enhanced cell proliferation cannot be definitely distinguished regarding the elevation of PE+/PE− rates. Primary cultured cells would be taken into consideration instead of cell lines in a further mechanism study to avoid unexpected cell proliferation.

## 5. Conclusions

In this study, we demonstrated that a reduction in CISD2 expression can exacerbate the inflammation and mitochondrial dysfunction associated with aging or injury-challenged neural status. Curcumin was shown to increase the expression of CISD2 in very old mice and neural cells with or without LPS-challenge. The beneficial effects of curcumin can be attributed to the preservation of CISD2. Protective properties against injury-induced inflammation and the mitochondrial dysfunction of curcumin are likely to be attenuated in cases of CISD2 deficiency.

## Figures and Tables

**Figure 1 nutrients-11-00700-f001:**
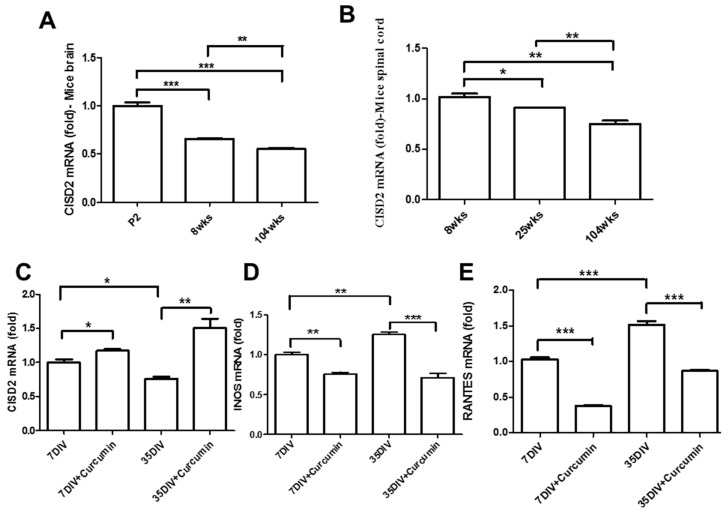
Curcumin upregulated CISD2 expression in aged mice in vivo. (**A**,**B**) In vivo mouse model of aging. Results of CISD2 mRNA expression in the brains (**A**) and spinal cords (**B**) of mice at P2, 8, 25 and 104 weeks (n = 4 in each group). Curcumin enhanced CISD2 expression in a long-term culture of neural cells in vitro. Results of mRNA expression of CISD2 (**C**), iNOS (**D**), and RANTES (**E**) in the neural cells of mice with or without 1 μM curcumin treatment at 7 *DIV* and 35 *DIV*. Vertical bars indicate the mean ± (standard error of the mean (SEM)) of mRNA expression (n = 3). * *p* < 0.05, ** *p* < 0.01, and *** *p* < 0.001 indicate differences of statistical significance.

**Figure 2 nutrients-11-00700-f002:**
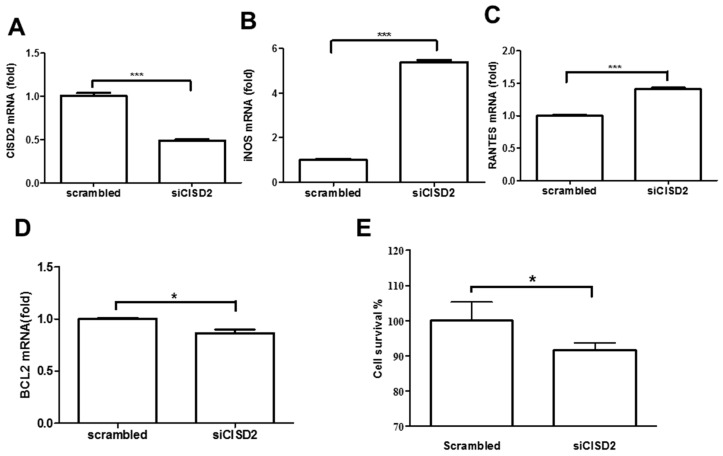
Knockdown of *CISD2* expression in neural cells produced inflammatory responses, promoted apoptosis, and decreased cell viability. Results of mRNA expression of CISD2 (**A**), iNOS (**B**), RANTES (**C**), and BCL2 (**D**) in neural cells with or without siCISD2 transfection. (**E**) Cell viability was measured using alamarBlue^®^ assay. Vertical bars indicate the mean ± (SEM) of mRNA expression (n = 3). * *p* < 0.05, and *** *p* < 0.001 indicate differences of statistical significance.

**Figure 3 nutrients-11-00700-f003:**
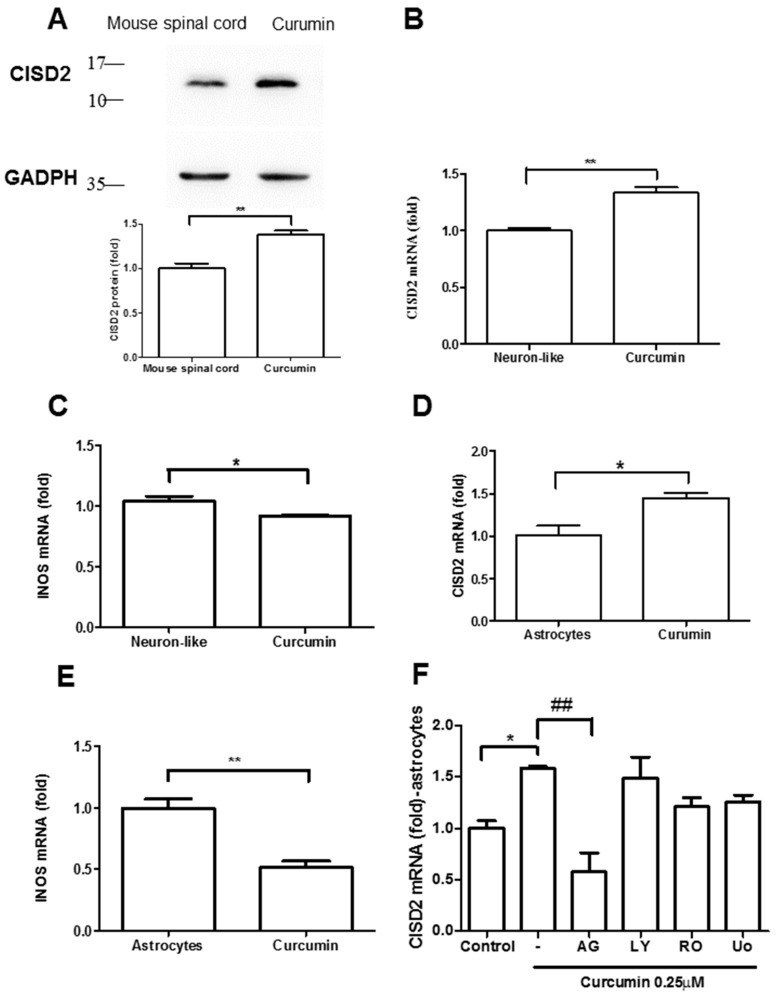
Curcumin upregulated CISD2 protein expression in vivo. (**A**) CISD2 protein expression in mouse spinal cords following intraperitoneal administration of curcumin at a concentration of 40 mg/kg. The upper panel presents the results of immunoblotting analysis of CISD2 (15 kDa) with GAPDH (37 kDa) serving as an internal control. The lower panel presents the mean ± (SEM) of CISD2/GAPDH band intensity as a ratio of the results from the control group (n = 4). Curcumin enhanced CISD2 mRNA expression in vitro via JAK/STAT signaling pathways. mRNA expression of CISD2 (**B**) and iNOS (**C**) in neuron-like cells (SH-SY5Y). mRNA expression levels of CISD2 (**D**) and iNOS (**E**) in astrocyte culture. (**F**) CISD2 mRNA expression following treatment with curcumin and the inhibition of JAK/STAT signaling pathways in non-stimulated astrocytes: AG490: JAK/STAT inhibitor, LY294002: PI3K inhibitor, RO318220: PKC inhibitor, U0126: MAPK inhibitor. Vertical bars indicate the mean + (SEM) of mRNA expression (n = 3). * *p* < 0.05, and ##,** *p* < 0.01 indicate differences of statistical significance.

**Figure 4 nutrients-11-00700-f004:**
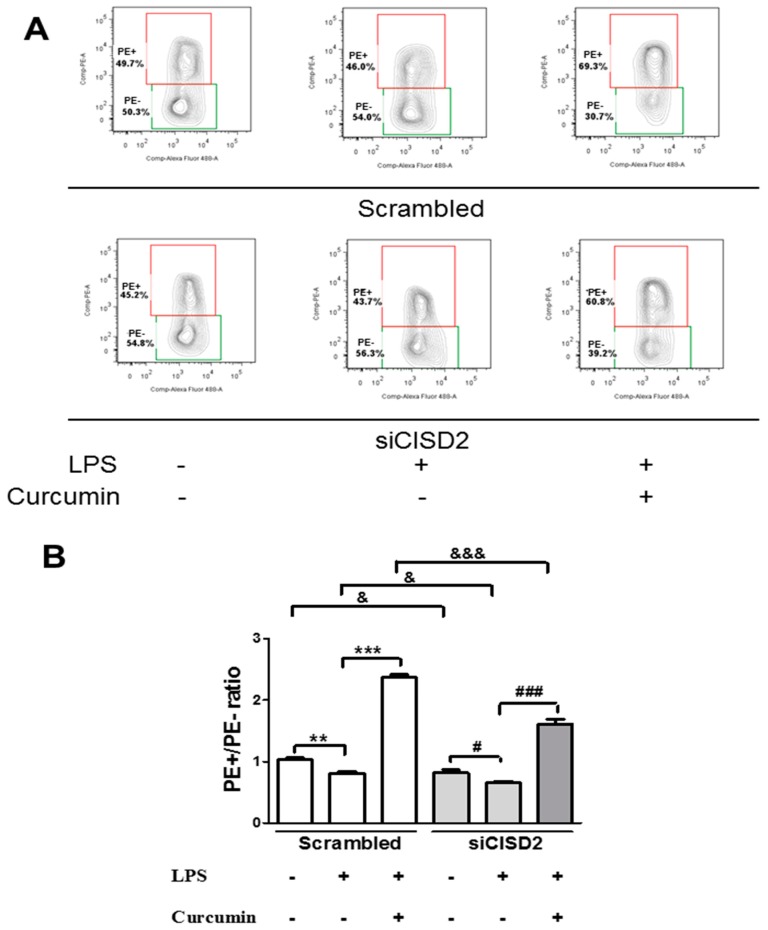
Mitochondrial membrane potential [DeltaPsi(m)] of lipopolysaccharide (LPS)-challenged neural cells (with or without curcumin treatment), as determined using JC-1 staining and flow cytometry with or without CISD2 knockdown. Cells presenting a decrease in DeltaPsi(m) and impending cellular apoptosis are indicated by monomer green JC-1 fluorescence, compared to normal cells with intact DeltaPsi(m) (indicated by j-aggregated red fluorescence). All groups of cells were evaluated in terms of the ratio of PE+ to PE−. (**A**) Representative results of flow cytometry for the following groups of cells: (i) scramble RNA-transfected control cells untreated with LPS; (ii) scramble RNA-transfected, LPS-challenged cells, untreated with curcumin; (iii) scramble RNA-transfected, LPS-challenged cells following curcumin treatment; (iv) siCID2-transfected control cells without LPS or curcumin treatment; (v) siCID2-transfected, LPS-challenged cells without curcumin treatment; (vi) siCID2-transfected, LPS-challenged cells, treated with curcumin. (**B**) Bars indicate the mean ± SEM (n = 3). #,& *p* < 0.05, ** *p* < 0.01, and ###,&&&,*** *p* < 0.001 indicate differences of statistical significance.

**Figure 5 nutrients-11-00700-f005:**
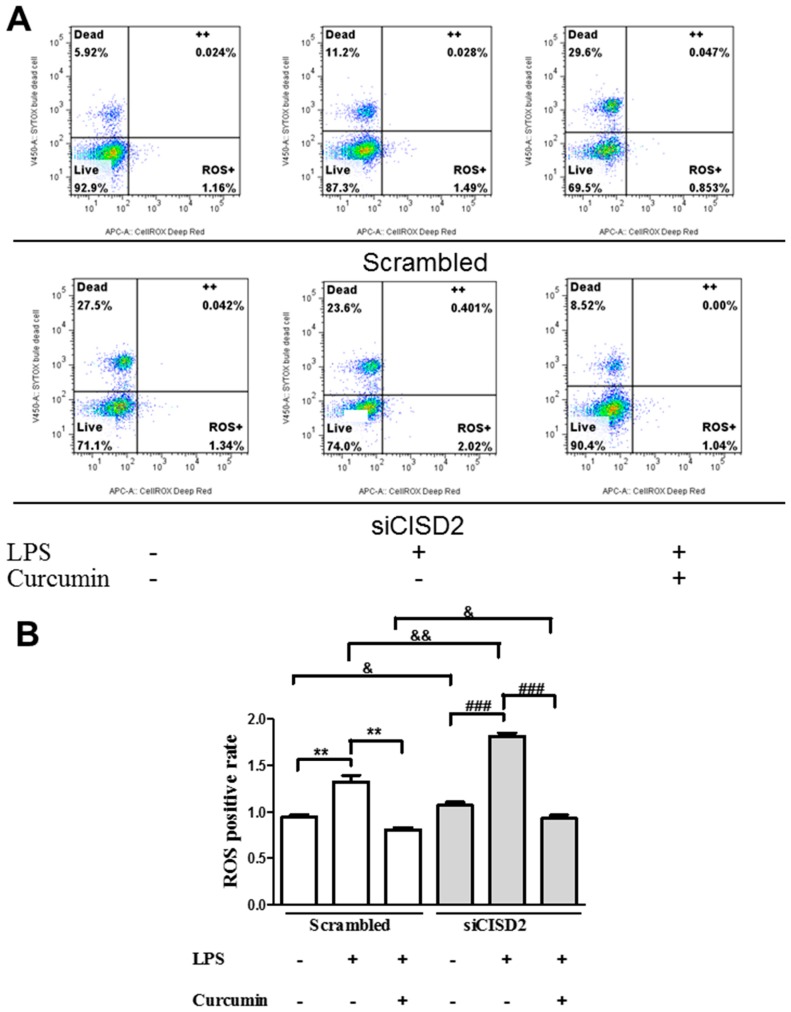
Reactive oxygen species (ROS) formation in LPS-challenged neural cells (with or without curcumin treatment), as determined by CellROX^®^ Deep Red/SYTOX^®^ Blue Dead Cell stain and flow cytometry with or without *CISD2* knockdown. RegionLive: vital cells; Region Dead: dead cells; Region++: dead cells with accumulated ROS; Region ROS+: vital cells with ROS formation. SYTOX ^®^ Blue Dead Cell stain was used to distinguish vital cells from dead cells. ROS+ cells of all groups were analyzed using flow cytometry (**A**). Representative flow cytometry results from all groups. (**B**) Bars indicate the mean ± SEM (n = 3). & *p* < 0.05, &&,** *p* < 0.01, and ### *p* < 0.001 indicate differences of statistical significance.

**Figure 6 nutrients-11-00700-f006:**
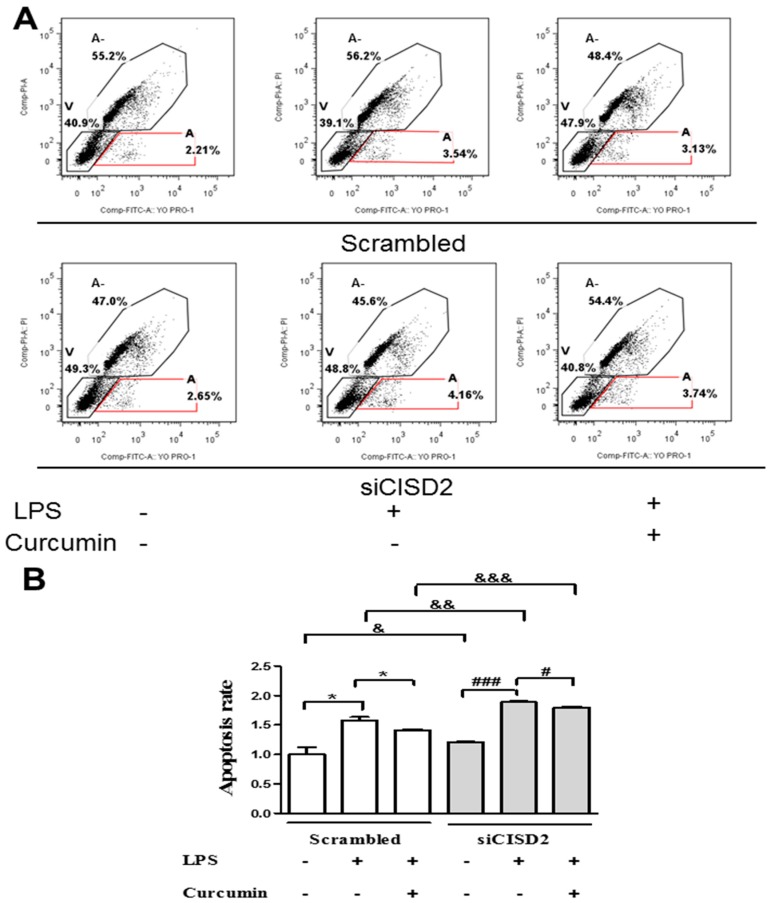
Apoptosis of LPS-challenged neural cells with or without curcumin treatment, as determined using YO-PRO-1/PI double staining and flow cytometry with or without CISD2 knockdown. YO-PRO-1 serves as a marker for apoptosis. PI refers to propidium iodide, which is used as a marker for necrotic cells. Region A-: non-living cells without early apoptosis; Region V: vital cells; Region A: early apoptotic cells. Early apoptotic cells with positive YO-PRO-1 and negative PI staining were calculated and compared among all groups (**A**). Representative flow cytometry results from all groups (**B**). Bars indicate the mean ± SEM (n = 3). &,#,* *p* < 0.05, && *p* < 0.01, and &&&,### *p* < 0.001 indicate differences of statistical significance.

**Figure 7 nutrients-11-00700-f007:**
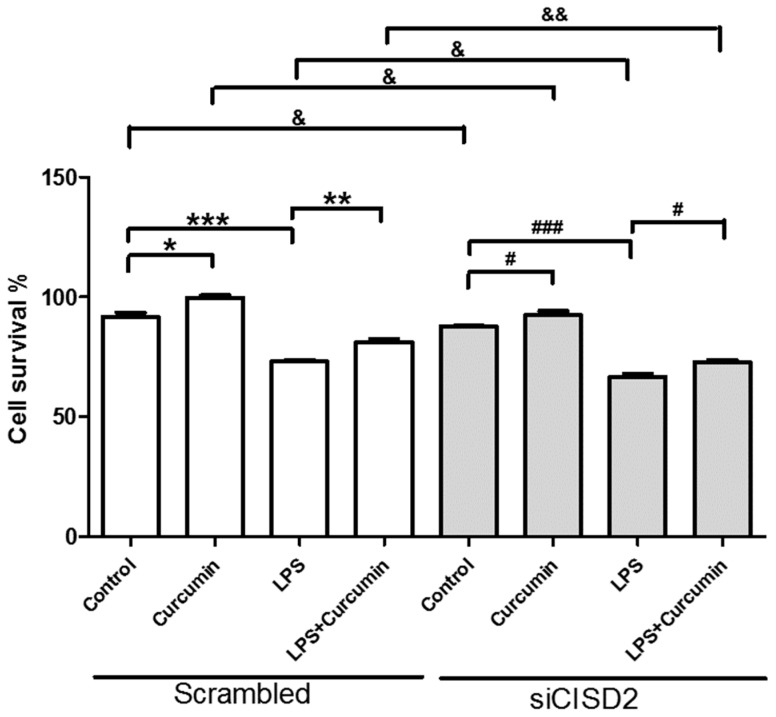
Curcumin enhanced cell viability and attenuated LPS-challenge-driven cell death in non-stressed and LPS-challenged neural cells. The knockdown of CISD2 expression was shown to reduce the protective effects of curcumin. The eight groups of cells included the following: (i) scrambled RNA-transfected control cells, untreated with LPS and curcumin; (ii) scrambled RNA-transfected cells treated with curcumin; (iii) scramble RNA-transfected, LPS-challenged cells, without curcumin treatment; (iv) scramble RNA-transfected, LPS-challenged cells treated with curcumin; (v) siCID2-transfected control cells without LPS or curcumin treatment; (vi) siCID2-transfected cells treated with curcumin; (vii) siCID2-transfected, LPS-challenged cells without curcumin treatment; (viii) siCID2-transfected, LPS-challenged cells, treated with curcumin. Cell viability was measured using alamarBlue^®^ assay. Vertical bars indicate the mean ± SEM (n = 3). #,&,* *p* < 0.05, &&,** *p* < 0.01, and ###,*** *p* < 0.001 indicate differences of statistical significance.

**Table 1 nutrients-11-00700-t001:** Primers used for real-time quantitative reverse-transcription polymerase chain reaction (qRT-PCR).

Gene	Orientation	Sequence
*CISD2*	forward	5′-AAAATCCCAAGGTGGTGAATGA-3′
	reverse	5′-GGACCGCCAGCACCTACA-3′
*Cyclophilin*	forward	5′-ATGGTCAACCCCACCGTGT-3′
	reverse	5′-CGTGTGAAGTCACCACCCT-3′
*BCL2*	forward	5′-TGGGATGCCTTTGTGGAACT-3′
	reverse	5′-CAGCCAGGAGAAATCAAACAGA-3′
*iNOS*	forward	5′-CCTCAGTTCTGCGCCTTTG-3′
	reverse	5′-GTTCGTCCCCTTCTCCTGTTG-3′
*RANTES*	forward	5′-TGCCCACGTCAAGGAGTATTT-3′
	reverse	5′-GGCGGTTCCTTCGAGTGA-3′

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
