# Peer review of "Protective Effects of CISD2 and Influence of Curcumin on CISD2 Expression in Aged Animals and Inflammatory Cell Model"

_nutrients, 2019, doi:10.3390/nu11030700_

Round 1

Reviewer 1 Report

The present manuscript by Lin C-C et al. demonstrated effects of pro-inflammatory factors and/or curcumin on CISD2 expression in neural cells. The experiments seem to be appropriately conducted and the manuscript is generally well written. However, similarly planned in vivo experiments and results appeared to be already published in ref. 19. In addition, the rationale for dose selection of a single concentration of 1 microM of curcumin in each in vitro experiment should be indicated, otherwise it is necessary to conduct additional experiments with dose responses should be added. Minor comments are listed below.

Line 138: The reason why was the curcumin administered to mice by intraperitoneal injection.

Line 162: The methods of establishment and culture of primary neural cells should be demonstrated.

Lines 266-271: Typing errors were found in the numbering of Figure 2.

Figure 4B: It is recommended to demonstrate the reason why the PE/A488 ratio of scrambled (LPS+Curcumin+) was clearly higher than (LPS-Curcumin-).

Author Response

The contents of Point-by-point response as attached file for reviewer 1  

Reviewer 2 Report

“Protective effects of CISD2 and influence of curcumin on CISD2 expression in aged animals and inflammatory cell model”

 Authors: Lin CC, Chiang TH, Sun YY, and Lin MS

Summary:

Impairment of CISD2 gene expression may stimulate mitochondrial dysfunction and inflammatory responses in neuronal cells. Curcumin has been shown to exert a protective role on mitochondrial function and inflammation in neuronal cells through stimulation of CISD2. Aging is connected with low-grade persistent chronic inflammation promoting neurological disorders and mitochondrial dysfunction, but the role of curcumin on CISD2 in aging is unknown. Therefore, the presented report investigated the potential protective role of curcumin on CISD2 expression in with LPS-treated neuronal cells in an in vivo and in vitro model of aging.

Comments:

As the overall world population is becoming older, treatment of problems associated with aging, especially neurological disorders become increasingly important.  Therefore, investigating novel compounds for targeting anti-inflammatory and anti-apoptotic protective signalling pathways in neuronal cells are interesting.

However, the presented study provides several major flaws:

1)    The Abstract is not concise enough. For example, line 25 and 32-33 should be excluded. Further, a clear structure, stating the aim of the study and results in a precise way is needed.

2)    Introduction: The authors have to make clear in the Introduction, that a) interaction of CISD2 and curcumin was already explored by their group (Lin CC et al. 2015) and b) that the novelty in this report is the impact of CISD2-curcumin interaction on aging of neuronal cells.

3)    In the introduction line 48-55, and 79-80 should be omitted.

4)    Introduction line 88: Please note that “curcumin” is not the whole plant, but only a chemically specific derivative from the root!

5)    Materials and Methods section is very poorly written! Important points to clarify: a) Please describe in detail exactly how each experiment was performed! b) When were human, mouse or rat cells used? c) When were astrocytes, primary neuronal cells or neuronal cell lines used? d) How were cells kept for long time cell culture of 35 days?

6)    For in vivo experiments, animals are intraperitoneally injected with curcumin. It would be very important to demonstrate that indeed, the observed effects on gene expression changes and mitochondrial function are related to action of curcumin on the neuronal cells. To do this, a) either show mechanism of how curcumin reaches CNS in animals or investigate levels of curcumin in the CNS specimen obtained from the animals. If this is not done, it is not plausible that observed in vivo effects are indeed induced by curcumin treatment. It could be for example due to hormonal or cytokine or other induction/influence in a complete other organ or cells.

7)    Results: Overall the results section is written in a confusing manner: Please a) describe separately in vivo and in vitro experiments and b) do not jump in the paper back and forth between the figure descriptions. Results and figures should be presented in a continuous way, starting description of results with the first figure and ending with the last figure. Additionally, figures should be presented in a clearly structured way. 

8)    It would be very important to show effects of curcumin on cell morphology and mitochondria with ultrastructure techniques! It is not sufficient to only demonstrate PCR and flow cytometry results.  

9)    Lines 387-393 belong to Materials and Methods section, not results part.

10) Labelling of figures has to be precise and represent labelling of figure legend and vice versa.

11) Quality of western blots presented in Fig. 3 is very poor.

12) Discussion does not adequately discuss obtained results and needs to focus more on the novelty of aging-CISD2-curcumin axis. Additionally, first part of Discussion is just repetition of results (lines 475-483).

13)  Lines 500-506: Do not start every sentence with “curcumin”.

Author Response

The contents of point-by-point response as attached file for reviewer 2

Reviewer 3 Report

The title of this manuscript indicates the authors seek to elucidate two aims in their study by using in vivo and in vitro models. While in the conclusion they only point out the protective effects of curcumin treatment which is partially attributed to increase CISD2 expression. Overall this is a preliminary study which is mostly relying on knocking-down conditions and with uneven quality of data presented.

1. The authors applied different time points in Fig.1A and 1B which is also inconstant with the legend. In Fig.1C and D, as the mRNA expression was indicated as "fold", should the mRNA expression of 7 DIV be normalized as 1?
2. The knockdown of CISD2 should be verified by not only the mRNA level but the protein level (by IB assay). Also I don't see any description of the method about cell survival.
3. The CISD2 panel of Fig.3A is spliced with uneven background which makes it difficult to compare. And again, the mRNA expression of control groups in 3E and 3F should be 1.
4.  In the Flow cytometry charts, the gates are different especially for the LPS+Curcumin groups. Also the percentages of A488 are mislabeling, because all the events are considered A488 (better as AF488) positive. The potential change can be measured as PE+/PE-. And based on the representative results, I don't think there is significant difference between the untreated groups.
5. In Fig. 5 and 6, the authors should present the representative results of all the groups as compared in bar graphs.
6. In Fig. 7, as read in the legend, the authors did the cell viability by alamarBlue assay which is not included in the methods section. I can see the cell viability of curcumin treatment is higher than control, especially in the scrambled group, which is higher than 100%. My question is how they calculate the percentage of cell survival? The curcumin treatment looks like can enhance cell proliferation. To evaluate the curcumin's effect on  LPS caused cell survival, should they use curcumin treated cell as normalization?

Additional issues.
1. The abstract is too wordy and should be concise.
2. The first paragraph in Introduction section should be deleted.

Author Response

The contents of point-by-point response as attached file for reviewer 3

Round 2

Reviewer 1 Report

Reviewer 1:

Comment: Re: Relationship between ref. 19 and the present manuscript.

> This reviewer understood the authors’ reply.

Comment: Re: The single concentration of curcumin

> This reviewer understood the authors’ reply.

Comment: Re: The reason why was the curcumin administered to mice by intraperitoneal injection.

> This reviewer understand that the purpose of the present study is to evaluate beneficial effects of orally ingested curcumin on CNS injuries or aging. Therefore, it is necessary to explain the biological equivalence of intraperitoneally injected and orally ingested curcumin. (Bibliographic consideration is acceptable.)

Comment: Re: Methods of establishment and cultured cells.

> This reviewer understood the authors’ reply.

Comment: Re: Typing errors were found in the numbering of Figure 2.

> This reviewer understood the authors’ reply.

Comment: Re: The reason why the PE/A488 ratio of scrambled (LPS+Curcumin+) was clearly higher than (LPS-Curcumin-) in Figure 4B.

> If 1 μM of curcumin enhanced proliferation of SH-SY5Y cells and PE/A488 ratios of (LPS+Curcumin+) group were elevated by the authors commented reactions, the results showed in Figure 4 could not be accepted. Because, the cause of the elevation of PE+/PE- rates of (LPS+Curcumin+) group was unclear whether apoptosis-related or not.

Author Response

Attached file includes answer to Reviewer 1

Reviewer 2 Report

“Protective effects of CISD2 and influence of curcumin on CISD2  expression in aged animals and inflammatory cell model”

 Authors: Lin CC, Chiang TH, Sun YY, and Lin MS

 Comments:

Overall, the majority of the questions raised by the reviewer have  been fully answered.1)    However, an important point to revise remains the  presentation of results. This should be in order of appearance!  For example “Results” from lines 244 to 283 all present Figure 1  and should be described as one block or in line 297 to 304 the  results in Figure 2 should be presented in a way that they can be  described from A to E without jumping back and forth. This goes  for all Figures! Figures should be re-adjusted to make it easier  and more logical for the reader to follow.2)    The authors have not given a plausible answer on how  curcumin reaches neuronal spinal cord cells and/or investigated  levels of curcumin in the CNS specimen obtained from the animals.3)    As in the Discussion they state in lines 662 to 666 that  favourable activity of curcumin is attributed to multi-targeting  potential, a potential pathway regulation of curcumin influencing  thereby CISD2 should be discussed!

Author Response

Attached file includes response to Reviewer 2

Reviewer 3 Report

Although the revised version has improvement, my major concern about the quality of data presented have not been well addressed, which might prevent its publication.

1. Fig.3A, the quality of IB is still low. Based on GAPDH, the samples were differently loaded which made the results not convincing.

2. From Fig.5A, which were newly added in the revised version, although the results of "Dead" portion were not analyzed in this study, these data still make me feel weird. In Scrambled group, curcumin plus LPS induced significant cell death, while in siCISD2, the result is the opposite. Also, the cell death with LPS treatment in siCISD2 is decreased!

3. In Fig.6A, which were also newly added in this revised version, we can see there are about 50% necrosis (or late apoptosis) cells with all treatments in both groups, how can this happen?

Author Response

Attached file includes response to Reviewer 3
